# The relevant resting-state brain activity of ecological microexpression recognition test (EMERT)

**Ming Yin[1]ͦ, Jianxin Zhang[2]ͦ\*, Deming Shu[3], Dianzhi Liu[3]\***

**1** Jiangsu Police Institute, Nanjing, China, **2** School of Humanities, Jiangnan University, Wuxi, China, **3** School of Education, Soochow University, Soochow, China

ͦ These authors contributed equally to this work.
\* 8201807147@jiangnan.edu.cn (JZ); psydzliu@163.com (DL)

**Data Availability Statement:** All relevant data are within the paper and its Supporting Information files.

**Funding:** This work was supported by "the Fundamental Research Funds for the Central Universities" of China (JUSRP11983, JZ, http://kjc.

## Abstract

Zhang, et al. (2017) established the ecological microexpression recognition test (EMERT), but it only used white models' expressions as microexpressions and backgrounds, and there was no research detecting its relevant brain activity. The current study used white, black and yellow models' expressions as microexpressions and backgrounds to improve the materials ecological validity of EMERT, and it used eyes-closed and eyes-open resting-state fMRI to detect relevant brain activity of EMERT for the first time. The results showed: (1) Two new recapitulative indexes of EMERT were adopted, such as microexpression $M$ and microexpression $SD$. The participants could effectively identify almost all the microexpressions, and each microexpression type had a significantly background effect. The EMERT had good retest reliability and calibration validity. (2) ALFFs (Amplitude of Low-Frequency Fluctuations) in both eyes-closed and eyes-open resting-states and ALFFs-difference could predict microexpression $M$. The relevant brain areas of microexpression $M$ were some frontal lobes, insula, cingulate cortex, hippocampus, parietal lobe, caudate nucleus, thalamus, amygdala, occipital lobe, fusiform, temporal lobe, cerebellum and vermis. (3) ALFFs in both eyes-closed and eyes-open resting-states and ALFFs-difference could predict microexpression $SD$, and the ALFFs-difference was more predictive. The relevant brain areas of microexpression $SD$ were some frontal lobes, insula, cingulate cortex, cuneus, amygdala, fusiform, occipital lobe, parietal lobe, precuneus, caudate lobe, putamen lobe, thalamus, temporal lobe, cerebellum and vermis. (4) There were many similarities and some differences in the relevant brain areas between microexpression $M$ and $SD$. All these brain areas can be trained to enhance ecological microexpression recognition ability.

## 1 Introduction

### 1.1 The ecological microexpression recognition test (EMERT)

Microexpressions are very transitory expressions lasting about 1/25–1/2 s, which can reveal people's true emotions they try to hide or suppress [1, 2]. Matsumoto et al. [3] developed the

jiangnan.edu.cn/), the Top-notch Academic
Programs Project of Jiangsu Higher Education
Institutions (PPZY2015A003, MY, http://jyt.jiangsu.
gov.cn/art/2015/6/1/art_38765_3245491.html),
the Mentoring Project of Philosophy and Social
Science Research in Colleges and Universities in
Jiangsu Province (2016): 2016SJD190004 (MY,
http://jyt.jiangsu.gov.cn/art/2016/6/28/art_58391_
7507970.html) and the National Natural Science
Foundation of China (31271084, DL, http://www.
nsfc.gov.cn/). JZ provided main research ideas and
financial support, and was responsible for research
design, data collection and analysis, and article
writing. MY provided part of research ideas and
financial support, and was responsible for research
design, data analysis, and article writing. DS was
responsible for experiment rogramming with E-
Prime and supporting experiment implementation.
DL was responsible for guiding the design,
implementation, data analysis and article writing of
the whole research, and provided financial support.

**Competing interests:** The authors have declared
that no competing interests exist.

Japanese and Caucasian Brief Affect Recognition Test (JACBART, classical microexpressions recognition) to measure microexpression recognition. The participants would see a microexpression presented for a little time between two neutral expression backgrounds for 2000ms before or after it. Participants needed to check out the microexpression type. The neutral expression backgrounds could eliminate the visual aftereffects of the microexpressions. But it did not examine the influence of backgrounds with emotional expressions. Therefore, Zhang, Fu, Chen and Fu [4] explored the background effect on microexpressions and found that all microexpressions (anger, disgust, fear, surprise and happiness) recognition accuracies under negative (sadness) backgrounds were significantly lower than those under positive (happiness) or neutral backgrounds; when the backgrounds and the microexpressions were consistent in the property (negative or positive), microexpression recognition accuracies were significantly lower than those when they were inconsistent in the property. The research has broken through the JACBART paradigm. But it did not explore all backgrounds or all microexpressions and needed to be further developed.

Yin, Zhang, Shi, and Liu [5] for the first time proposed that all basic expression kinds for both backgrounds and microexpressions needed to be detected to set up the ecological microexpression recognition test. Therefore, Zhang et al. [6] examined the recognition characteristics of six basic expression kinds of microexpressions (sadness, fear, anger, disgust, surprise, happiness) under seven basic expression kinds of backgrounds (the six basic expressions and neutral) to establish an ecological microexpression recognition test—EMERT, and found that EMERT had good retest reliability, criterion validity and ecological validity: (1) EMERT was generally significantly related to JACBART. (2) The backgrounds main effect of sadness, fear, anger and disgust microexpressions were significant; the backgrounds main effect of surprise and happiness microexpressions were not significant, but there was a wide difference between them with the common expressions. (3) The ecological microexpression recognition had stable fluctuation. Zhu et al. [7] used the simplified edition of EMERT to find microexpression recognition difference between depressive patients and normal people. Yin, Tian, Hua, Zhang, and Liu [8] extended EMERT to WEMERT (weak ecological microexpression recognition test).

But EMERT by Zhang et al. [6] only used white models' expressions as microexpressions and backgrounds, so white, black and yellow models' expressions need to be used as microexpressions and backgrounds to improve the ecological validity of materials.

## 1.2 Brain activation of ecological microexpression recognition

There were few published types of research detecting brain activation of ecological microexpression recognition. Shen [9] in Xiaolan Fu's team used fNIRS to find that the brain area responsible for JACBART microexpressions recognition was in the left frontal lobe, while the brain area responsible for common expressions recognition was in the right frontal lobe. Zhang [10] in Xiaolan Fu's team used fMRI to find that for anger and neutral microexpressions, the inferior parietal lobule was activated more in the negative expression backgrounds than in the neutral expression backgrounds, while the right precuneus was activated more in the positive expression backgrounds than in the neutral expression backgrounds. For happiness microexpressions, the parahippocampal gyrus was activated more in positive backgrounds. These studies revealed the brain mechanisms of classical microexpressions and three ecological microexpression recognition, but more ecological microexpression recognition needs further research.

As there were 36 ecological microexpressions in EMERT [6], it is neither feasible nor economical to adopt task-state fMRI. Resting-state fMRI is a viable and economical option. Resting-state fMRI investigates spontaneous activity or functional connections within the brain at

rest. If a certain cognitive task is associated with certain brain areas that are active in a resting state, then these brain areas are associated with the cognitive task. If brain areas whose activity in resting state related to two cognitive tasks differ, then the brain mechanisms underlying execution of these two cognitive tasks are different [11–13]. Brain spontaneous activity in resting-state is a stable index to measure the individual cognitive characteristics [14]. One of the classic indexes is ALFFs value (the Amplitude of Low-Frequency Fluctuations, 0.01 ~ 0.1 HZ), including most of the psychological, cognitive process. The higher and lower amplitudes are background noise such as physiological activity. There were eyes-closed and eyes-open resting-states. Nakano, etc. [15, 16] found that in eyes-closed, subjects focused on internal feeling and self-consciousness, while in eyes-open, subjects turned to external stimulus processing, and the transition from eyes-closed to eyes-open was from internal feeling and self-consciousness to external stimulus processing. However, there was no microexpression research using resting-state fMRI.

### 1.3 Improvements made in the current study

The current study would use white, black and yellow models' expressions as microexpressions and backgrounds to improve the materials ecological validity of EMERT. It would use eyes-closed and eyes-open resting-state fMRI to detect relevant resting-state brain activity of EMERT for the first time.

## 2 Methods

### 2.1 Participants

Sixty-five college students were selected to participate in the study. Males and females were 32 and 33. The age $M \pm SD$ = 21.71 ± 2.58. They were all right-handed with normal or corrected-to-normal eyesight and without colour blindness. They all volunteered and could quit at any time. Each participant completed an informed consent form before the experiments. They got corresponding rewards after completing the experiments. The experiments were by the ethical guidelines of the Declaration of Helsinki and were approved by the Scientific Review Committee of Faculty of Psychology, Southwest University, China.

### 2.2 Experimental apparatus and materials

Seven kinds of basic expression opened mouth pictures of eight models (four male and four female, including white, black and yellow people) from the NimStim face expression database [17] were used as the backgrounds, namely, neutral, sadness, fear, anger, disgust, surprise, and happiness. Except for neutral expression, the other six kinds of expressions were used as microexpressions. The pixels of all images were modified to be 338 × 434 with grey background (GRB: 127, 127, 127) [6]. A custom experimental program ran under E-prime 2.0 on a PC (Lenovo LX-GJ556D), with a 17-inch colour display (resolution 1024 × 768, refresh rate 60 Hz).

### 2.3 Experimental design and procedures

The experiment was 7 (expression backgrounds: neutral vs sadness vs. fear vs anger vs disgust vs surprise vs happiness) × 6 (microexpressions: sadness vs fear vs. anger vs disgust vs surprise vs happiness) × 2(test times: first EMERT vs the second EMERT) within-subject design. As there were seven kinds of expression backgrounds, to balance the sequential effect, the Latin square design was used to set up seven groups with about nine participants (four or five

females and males) in each group. Each dependent variable in seven groups was averaged in the result analysis [6].

Participants were 70 cm away from the screen. On the computer keyboard, six keys of SDF-JKL corresponded with 'anger', 'disgust', 'fear', 'sadness', 'surprise' and 'happiness'. Before the experiment, the participants were asked to put the ring finger, middle finger, index finger of their left hands on the SDF keys respectively while the index finger, middle finger, ring finger of their right hands on JKL keys. And then they did key pressing practices. First, one of the six kinds of expressions (except neutral) was presented 1000 ms; then six labels "anger, disgust, fear, sadness, surprise, happiness" appeared on the screen, and the participants needed to recognise it and press the right key as accurately as possible. There were 30 trials, and six kinds of expressions were pseudo-randomly presented for 5 times.

After the key pressing practice was completed, the instructor informed the participants of the procedure. First, the centre of the screen would show the "+" for 400 ms; second, the empty screen lasted 200 ms; then the front background expression image was presented for 800 ms, after which the microexpression image would appear for 133 ms, followed by 800 ms of back background expression image [3, 6]. The front and back backgrounds and microexpressions were of the same model's face, and the front and back backgrounds were the same. Participants needed to try to identify the briefly-presented microexpression between front and back backgrounds. Later, six labels "anger, disgust, fear, sadness, surprise, happiness" appeared on the screen. The participants were asked to press a key according to the microexpression they saw as accurately as possible instead of as soon as possible (no time limit). After the participants pressed the key, an empty screen would show for 1000 ms. Then the fixation point "+" was presented for 400 ms, and the next trial started. The experiment procedure is shown in Fig 1.

The participants practised the experimental procedure after understanding the instructions. There was a total of 14 trials, of which 7 kinds of backgrounds appeared 2 times, and 6 kinds of microexpressions each appeared 2 to 3 times. The participants were asked to determine the type of microexpressions. After the experimental procedure practice was completed, they started a formal trial. To allow the participants to get enough rest, the experiment was divided into seven blocks. Rest between every two blocks was 1 minute. The experiment had 7 (backgrounds) × 6 (microexpressions) × 8 (models) = 336 trails.

The participants needed to do two EMERT measurements, of which interval time was at least one day.

## 2.4 Resting-state data collection and analysis

The fMRI data were collected using a Siemens 3.0 T magnetic resonance imaging scanner, and an 8-channel phased front head coil. Eyes-closed and eyes-open resting-state imaging used gradient echo (GRE) single-excitation echo-planar imaging (EPI). Scan parameters were as follows: TR = 2000 ms, TE = 30 ms, FA = 90˚, FOV = $220 \times 220$ mm$^2$, matrix size = $64 \times 64$ mm$^2$,

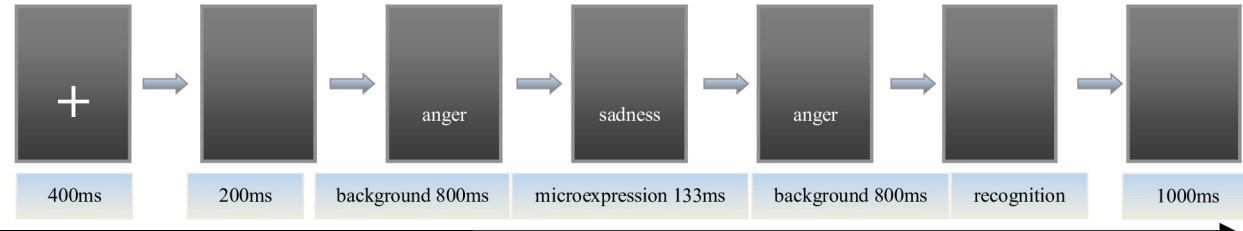

**Fig 1. The picture of experiment procedure.** Note: These images are licensed by the copyright owner, Tottenham et al [17].

depth = 3 mm, planar resolution = $3.13 \times 3.13$ mm$^2$, interval scanning, 33 layers, layer spacing = 0.6 mm, total 240 layers. Structural imaging used a 3D TlWI (MP-RAGE) sequence with sagittal scans. Scan parameters were the following: TR = 2600 ms, TE = 3.02 ms, FA = 8˚, no interval, FOV = $256 \times 256$ mm$^2$, matrix size = $256 \times 256$ mm$^2$, total 176 layers. All the participants first received the structural scan, then half received the eyes-closed and eyes-open resting-state scans, and half received the eyes-open and eyes-closed resting-state scans.

Pretreatment and analysis of resting-state data used DPARSF 3.0 Advanced Edition Calculate [18] in Original Space (Warp by DARTEL), following standard procedures: (1) Conversion of raw DICOM-format data to NIFTI format. To allow for signal stabilisation of the image, the first 10 TR images were removed, after which time layer correction (slice timing) and head motion correction (realign) were conducted. If head motion greater than 2 mm occurred during resting-state, the data were deleted. (2) The new segment + DARTEL was used to split the structural T1 data without standardisation, and register the T1 split data directly to the resting-state functional images. Before registration of structural and functional data, the AC-PC line of each participant's T1 image and the resting-state function was registered, and then automatic registration was applied. Therefore, the resting-state analysis took place in the original T1 space. (3) Head motion (adopting Friston 24), linear drift, white matter, and cerebrospinal fluid via regression were adjusted for. (4) Low frequency fluctuations ALFFs (filter range: 0.01 to 0.1 Hz) were calculated. (5) The resting-state function was registered to the standard MNI space (normalisation), using a $3 \times 3 \times 3$ mm$^3$ voxel size, with $4 \times 4 \times 4$ mm$^3$ full widths at half maximum (FWHM) smoothing.

REST1.8 [19] was first used to extract the amplitude of low-frequency fluctuations (ALFFs) during resting-states in 116 Anatomical Automatic Labeling (AAL) brain areas. Second, SPSS19.0 was used to implement correlation analyses between ALFFs in 116 AAL brain areas and the scores of EMERT. The ALFFs-difference of eyes-open minus eyes-closed was used as an index of transition from internal feeling and self-consciousness to external stimulus processing. We detected its psychological significance by correlation analyses between it and the scores of EMERT.

## 3 Results

SPSS 19.0 was used for statistics. There were sixty-five valid participants in EMERT, fifty-eight valid participants in eyes-closed resting-state, and sixty two valid participants in eyes-open resting-state because seven participants' head movement were greater than 2 mm in eyes-closed resting-state, and three participants in eyes-open resting-state.

### 3.1 Behavioral data

The scores of EMERT were showed in Table 1. Because the accuracy of microexpression recognition in the second EMERT might contain training effect, the accuracy of that in the first EMERT was taken as the microexpression recognition ability. Since the participants have 6 keys to choose for each trial, the random level is 1/6. A single sample *t* test was made for each microexpression recognition accuracy with random level 1/6, and it was found that almost all the microexpression recognition accuracies in the first EMERT were significantly higher than random ($p$s<0.001), except that fear under surprise, was not significantly higher than random ($p$>0.05).

It can be seen that the EMERT indexes were too many to be recapitulative enough for both participants and researchers. Then the mean of accuracy rates of a microexpression type under six backgrounds (except the same expression grounds as the microexpression, because in that case, it was a normal expression rather than a microexpression) was used as the index of this

**Table 1. The scores of EMERT.**

| microexpressions | first EMERT $M\pm SD$ ($n = 65$) | $t$ | Cohen's $d$ | second EMERT $M\pm SD$ ($n = 65$) |
|---|---|---|---|---|
| sadness | 0.60±0.31 | 11.04*** | 1.40 | 0.74±0.27 |
| fear | 0.50±0.32 | 8.35*** | 1.04 | 0.57±0.33 |
| anger | 0.68±0.28 | 14.47*** | 1.83 | 0.78±0.25 |
| disgust | 0.60±0.32 | 10.87*** | 1.35 | 0.74±0.31 |
| surprise | 0.78±0.27 | 18.18*** | 2.27 | 0.78±0.31 |
| happiness | 0.90±0.24 | 24.35*** | 3.06 | 0.92±0.23 |
| sadness under fear | 0.29±0.24 | 4.20*** | 0.51 | 0.39±0.29 |
| sadness under anger | 0.34±0.28 | 4.87*** | 0.62 | 0.44±0.30 |
| sadness under disgust | 0.28±0.22 | 4.15*** | 0.52 | 0.37±0.25 |
| sadness under neutral | 0.38±0.24 | 7.26*** | 0.89 | 0.43±0.25 |
| sadness under surprise | 0.47±0.28 | 8.84*** | 1.08 | 0.56±0.30 |
| sadness under happiness | 0.34±0.25 | 5.84*** | 0.69 | 0.43±0.27 |
| fear under sadness | 0.28±0.18 | 4.99*** | 0.63 | 0.28±0.23 |
| fear under anger | 0.27±0.20 | 4.25*** | 0.52 | 0.30±0.21 |
| fear under disgust | 0.29±0.21 | 4.76*** | 0.59 | 0.38±0.26 |
| fear under neutral | 0.34±0.21 | 6.93*** | 0.83 | 0.36±0.25 |
| fear under surprise | 0.18±0.20 | 0.46 | - | 0.19±0.21 |
| fear under happiness | 0.44±0.25 | 9.08*** | 1.09 | 0.39±0.26 |
| anger under sadness | 0.72±0.25 | 17.96*** | 2.21 | 0.76±0.24 |
| anger under fear | 0.70±0.27 | 15.97*** | 1.98 | 0.72±0.26 |
| anger under disgust | 0.49±0.29 | 9.24*** | 1.11 | 0.55±0.29 |
| anger under neutral | 0.76±0.23 | 20.65*** | 2.58 | 0.75±0.24 |
| anger under surprise | 0.70±0.29 | 14.64*** | 1.84 | 0.73±0.30 |
| anger under happiness | 0.62±0.26 | 13.88*** | 1.74 | 0.66±0.26 |
| disgust under sadness | 0.47±0.23 | 10.93*** | 1.32 | 0.56±0.25 |
| disgust under fear | 0.55±0.27 | 11.60*** | 1.42 | 0.59±0.27 |
| disgust under anger | 0.40±0.26 | 7.46*** | 0.90 | 0.54±0.28 |
| disgust under neutral | 0.54±0.27 | 10.97*** | 1.38 | 0.62±0.25 |
| disgust under surprise | 0.44±0.23 | 9.54*** | 1.19 | 0.49±0.28 |
| disgust under happiness | 0.49±0.25 | 10.40*** | 1.29 | 0.59±0.24 |
| surprise under sadness | 0.67±0.24 | 16.93*** | 2.10 | 0.66±0.25 |
| surprise under fear | 0.73±0.25 | 18.37*** | 2.25 | 0.72±0.29 |
| surprise under anger | 0.67±0.29 | 14.21*** | 1.74 | 0.61±0.30 |
| surprise under disgust | 0.65±0.28 | 13.82*** | 1.73 | 0.62±0.28 |
| surprise under neutral | 0.78±0.24 | 20.38*** | 2.56 | 0.80±0.25 |
| surprise under happiness | 0.72±0.30 | 14.80*** | 1.84 | 0.72±0.29 |
| happiness under sadness | 0.88±0.25 | 22.89*** | 2.85 | 0.90±0.21 |
| happiness under fear | 0.86±0.25 | 22.26*** | 2.77 | 0.89±0.23 |
| happiness under anger | 0.85±0.29 | 19.16*** | 2.36 | 0.88±0.25 |
| happiness under disgust | 0.83±0.29 | 18.72*** | 2.29 | 0.87±0.25 |
| happiness under neutral | 0.91±0.24 | 24.98*** | 3.10 | 0.95±0.15 |
| happiness under surprise | 0.85±0.28 | 19.30*** | 2.44 | 0.90±0.24 |

Note:

* $p < 0.05$

** $p < 0.01$

*** $p < 0.001$. The same below.

microexpression type recognition. It was abbreviated as microexpression *M*. The standard deviation of accuracy rates of this microexpression type under six backgrounds (except the same expression grounds as the microexpression) was used as the background effect index of this microexpression type recognition, which was called the fluctuations of the microexpression type recognition [6, 8], and it was abbreviated as microexpression *SD*.

Therefore we got two new recapitulative indexes of EMERT. A single sample *t* test was made for each microexpression *M* with random level 1/6, and it was found that all were significantly higher than random (*p*s<0.001). A single sample *t* test was made for each microexpression *SD* with random level 0, and it was found that all were significantly higher than random (*p*s<0.001).

Pearson correlation was made between the two EMERT. It was found that each microexpression *M* in the first EMERT was significantly positively related to the corresponding one in the second EMERT and the *r*s (the plural of *r*, the same below) were high; and that each microexpression *SD* except surprise *SD* in the first EMERT was significantly positively related to the corresponding one in the second EMERT.

Pearson correlation was made between the first EMERT and the first JACBART (microexpressions under neutral backgrounds belong to the first EMERT), and it was found that each microexpression *M* in the first EMERT was significantly positively related to the corresponding microexpression in the first JACBART. The new indexes of the two EMERT and their statistical results were shown in Table 2.

## 3.2 Brain imaging data

Pearson correlation analysis was made between ALFFs of resting-state and microexpression *M* (see Table 3 and Fig 2). (1) In the eyes-closed resting state, ALFFs in the frontal lobe, insula, cingulate cortex, hippocampal, caudate nucleus, thalamus and vermis were significantly correlated with some microexpression *M*. (2) In the eyes-open resting state, ALFFs in the frontal lobe, insula, cingulate cortex, hippocampus, parietal lobe, caudate nucleus, thalamus, temporal lobe, cerebellum and vermis were significantly correlated with some microexpression *M*. (3) In the ALFFs-difference of eyes-open minus eyes-closed resting-states, ALFFs-difference in the frontal lobe, insula, amygdala, occipital lobe, fusiform, temporal lobe, cerebellum and vermis were significantly correlated with some microexpression *M*.

**Table 2. The new scores of EMERT.**

| microexpressions | first EMERT M±SD (n = 53) | t | Cohen's d | second EMERT M±SD (n = 53) | $r_1$ | first JACBART M±SD (n = 53) | $r_2$ |
|---|---|---|---|---|---|---|---|
| sadness M | 0.35±0.20 | 7.59*** | 0.92 | 0.43±0.24 | 0.79** | 0.38±0.24 | 0.68** |
| fear M | 0.30±0.13 | 8.07*** | 1.03 | 0.32±0.18 | 0.66** | 0.34±0.21 | 0.52** |
| anger M | 0.67±0.22 | 18.19*** | 2.29 | 0.69±0.22 | 0.83** | 0.76±0.23 | 0.77** |
| disgust M | 0.48±0.21 | 12.13*** | 1.49 | 0.56±0.23 | 0.82** | 0.54±0.27 | 0.90** |
| surprise M | 0.70±0.22 | 19.98*** | 2.42 | 0.69±0.24 | 0.79** | 0.78±0.24 | 0.79** |
| happiness M | 0.86±0.24 | 23.32*** | 2.89 | 0.90±0.20 | 0.91** | 0.91±0.24 | 0.81** |
| sadness SD | 0.17±0.07 | 18.59*** | 2.43 | 0.16±0.07 | 0.43** | | |
| fear SD | 0.18±0.07 | 20.78*** | 2.57 | 0.17±0.06 | 0.41** | | |
| anger SD | 0.17±0.07 | 19.21*** | 2.43 | 0.15±0.09 | 0.55** | | |
| disgust SD | 0.15±0.06 | 19.20*** | 2.50 | 0.13±0.06 | 0.26* | | |
| surprise SD | 0.16±0.08 | 15.48*** | 3.20 | 0.15±0.07 | - | | |
| happiness SD | 0.08±0.1 | 6.95*** | 0.80 | 0.07±0.09 | 0.68** | | |

Note: $r_1$ was the *r* between first and second EMERT. $r_2$ was the *r* between first EMERT and first JACBART.

**Table 3. The *r*s between ALFFs of resting-state and microexpression *M*.**

| resting-state | AAL brain area | ALFF (*M*±*SD*) | sadness *M* | fear *M* | anger *M* | disgust *M* | surprise *M* | happiness *M* |
|---|---|---|---|---|---|---|---|---|
| eyes-closed | Precentral_L | 0.84±0.05 | | | | -0.39** | | |
| eyes-closed | Precentral_R | 0.88±0.07 | | | | -0.294* | | |
| eyes-closed | Frontal_Inf_Tri_R | 0.78±0.04 | | | | -0.31* | | |
| eyes-closed | Frontal_Inf_Orb_L | 0.88±0.06 | | | | 0.31* | | |
| eyes-closed | Rolandic_Oper_R | 0.86±0.03 | 0.27* | 0.30* | | | | |
| eyes-closed | Insula_L | 0.91±0.04 | | | | 0.28* | | |
| eyes-closed | Insula_R | 0.96±0.05 | | | | 0.28* | | |
| eyes-closed | Cingulum_Ant_L | 0.98±0.06 | 0.27* | | | | | |
| eyes-closed | Cingulum_Mid_L | 0.94±0.03 | 0.33* | 0.29* | 0.30* | | | |
| eyes-closed | Cingulum_Mid_R | 0.92±0.03 | 0.33* | | 0.30* | | | |
| eyes-closed | Cingulum_Post_L | 0.98±0.05 | 0.26* | | | 0.262* | | |
| eyes-closed | Cingulum_Post_R | 0.93±0.04 | 0.30* | | | | | |
| eyes-closed | ParaHippocampal_L | 1.11±0.09 | | 0.27* | | | | |
| eyes-closed | Caudate_R | 0.86±0.05 | | | -0.28* | | -0.27* | |
| eyes-closed | Thalamus_L | 1.01±0.09 | | | -0.29* | | -0.36** | -0.37** |
| eyes-closed | Thalamus_R | 1.01±0.08 | | -0.26* | -0.39** | | -0.45** | -0.39** |
| eyes-closed | Vermis_6 | 0.94±0.07 | | | | | | -0.29* |
| eyes-closed | Vermis_7 | 0.81±0.07 | | | -0.32* | | -0.26* | -0.41** |
| eyes-closed | Vermis_10 | 2.28±0.67 | | | -0.29* | | -0.29* | |
| eyes-open | Precentral_L | 0.82±0.04 | | | | -0.38** | | |
| eyes-open | Precentral_R | 0.85±0.05 | | | | -0.28* | | |
| eyes-open | Frontal_Sup_L | 0.87±0.05 | | | 0.27* | | | |
| eyes-open | Frontal_Sup_Orb_R | 0.83±0.07 | | -0.33** | | | | |
| eyes-open | Frontal_Inf_Oper_R | 0.89±0.04 | | | 0.26* | | | |
| eyes-open | Frontal_Inf_Orb_L | 0.89±0.05 | | | | 0.31* | 0.27* | |
| eyes-open | Frontal_Inf_Orb_R | 0.81±0.04 | 0.25* | | 0.27* | | 0.28* | |
| eyes-open | Rolandic_Oper_R | 0.86±0.03 | 0.37** | 0.39** | 0.27* | | | 0.26* |
| eyes-open | Frontal_Sup_Medial_L | 0.97±0.07 | 0.31* | | 0.31* | 0.25* | 0.32* | 0.32* |
| eyes-open | Frontal_Sup_Medial_R | 0.94±0.06 | | | | | 0.27* | |
| eyes-open | Insula_L | 0.92±0.03 | | | | 0.34** | | |
| eyes-open | Insula_R | 0.97±0.04 | 0.30* | | 0.34** | 0.31* | 0.25* | |
| eyes-open | Cingulum_Ant_L | 0.99±0.06 | 0.27* | 0.26* | 0.29* | 0.31* | | |
| eyes-open | Cingulum_Mid_L | 0.93±0.03 | 0.35** | 0.30* | | | | |
| eyes-open | Cingulum_Post_L | 1±0.05 | | 0.32* | 0.26* | 0.32* | 0.35** | |
| eyes-open | Cingulum_Post_R | 0.94±0.03 | | | | | 0.33** | |
| eyes-open | ParaHippocampal_L | 1.12±0.08 | | 0.33** | | | | |
| eyes-open | Postcentral_R | 0.84±0.05 | | | | -0.30* | | |
| eyes-open | Parietal_Inf_R | 1.06±0.06 | | | 0.28* | | | |
| eyes-open | Caudate_R | 0.87±0.05 | | | -0.25* | | | |
| eyes-open | Thalamus_L | 1±0.08 | | | -0.29* | | -0.32* | -0.33** |
| eyes-open | Thalamus_R | 1±0.07 | | -0.29* | -0.42** | | -0.44** | -0.40** |
| eyes-open | Temporal_Mid_R | 0.98±0.03 | | 0.25* | | | | |
| eyes-open | Cerebelum_Crus1_L | 0.95±0.09 | | | -0.31* | | | -0.37** |
| eyes-open | Cerebelum_Crus2_L | 0.63±0.22 | | | | | | -0.29* |
| eyes-open | Cerebelum_9_R | 0.79±0.28 | | -0.32* | | | | |
| eyes-open | Vermis_7 | 0.82±0.06 | | | | | | -0.29* |
| eyes-open | Vermis_10 | 2.25±0.68 | | | -0.31* | | -0.30* | -0.26* |

(*Continued*)

**Table 3.** (Continued)

| resting-state | AAL brain area | ALFF (M±SD) | sadness M | fear M | anger M | disgust M | surprise M | happiness M |
|---|---|---|---|---|---|---|---|---|
| difference | Supp_Motor_Area_L | -0.02±0.04 | | | | | | 0.30* |
| difference | Olfactory_L | 0.01±0.03 | | | | | 0.32* | 0.40** |
| difference | Insula_R | 0.01±0.02 | | | | | 0.29* | 0.38** |
| difference | Amygdala_R | 0.01±0.04 | 0.27* | | | | | |
| difference | Calcarine_L | -0.08±0.1 | | | | | | -0.29* |
| difference | Calcarine_R | -0.07±0.08 | | | | | | -0.30* |
| difference | Lingual_L | -0.06±0.08 | | | | | | -0.32* |
| difference | Occipital_Sup_R | 0±0.05 | | | | | | -0.32* |
| difference | Occipital_Mid_L | 0.01±0.04 | | | | | | -0.30* |
| difference | Occipital_Inf_L | 0.01±0.05 | | | | | | -0.34** |
| difference | Fusiform_L | 0.01±0.02 | | | | | | -0.27* |
| difference | Heschl_R | -0.04±0.06 | | | | | 0.34** | |
| difference | Temporal_Pole_Sup_L | 0.01±0.05 | 0.33* | | | | 0.32* | 0.30* |
| difference | Temporal_Pole_Sup_R | 0±0.05 | | | 0.34* | 0.28* | 0.33* | 0.32* |
| difference | Temporal_Mid_L | 0±0.02 | | 0.31* | | 0.28* | | |
| difference | Temporal_Pole_Mid_R | 0±0.03 | | | | 0.28* | | |
| difference | Cerebelum_Crus1_L | -0.01±0.05 | | | -0.27* | | | -0.31* |
| difference | Cerebelum_Crus2_L | -0.01±0.09 | -0.29* | | -0.27* | | | |
| difference | Cerebelum_6_L | -0.02±0.05 | | | | | | -0.34* |
| difference | Cerebelum_7b_L | -0.01±0.09 | | | -0.28* | -0.30* | | |
| difference | Cerebelum_7b_R | 0.01±0.1 | | | | -0.26* | | |
| difference | Cerebelum_8_L | 0±0.11 | | | | -0.31* | | |
| difference | Vermis_6 | -0.01±0.04 | | 0.26* | | | | |
| difference | Vermis_7 | 0.01±0.03 | | 0.28* | 0.31* | | | 0.30* |
| difference | Vermis_9 | -0.01±0.09 | | | -0.29* | | | |

Pearson correlation analysis was made between ALFFs of resting-state and microexpression *SD* (see Table 4 and Fig 3). (1) In the eyes-closed resting-state, ALFFs in the frontal lobe, insula, cingulate cortex, occipital lobe, parietal lobe, precuneus, caudate lobe, putamen lobe, thalamus, temporal lobe, cerebellum and vermis were significantly correlated with some microexpression *SD*. (2) In the eyes-open resting-state, ALFFs in the frontal lobe, insula, cingulate cortex, cuneus, occipital lobe, parietal lobe, precuneus, caudate lobe, putamen lobe, thalamus, temporal lobe, cerebellum and vermis were significantly correlated with some microexpression *SD*. (3) In the ALFFs-difference of eyes-open minus eyes-closed resting-states, ALFFs-difference in the frontal lobe, insula, cingulate cortex, amygdala, fusiform, occipital lobe, parietal lobe, temporal lobe, cerebellum and vermis were significantly correlated with some microexpression *SD*.

## 4 Discussion

### 4.1 The EMERT had good reliability and validity

In the current study, we used white, black and yellow models' expressions as microexpressions and backgrounds to improve the materials ecological validity of EMERT, and used two new recapitulative indexes such as microexpression *M* and microexpression *SD*.

Almost all the microexpression recognition accuracies and all the microexpression *M*s were significantly higher than random, which means that the participants could effectively identify

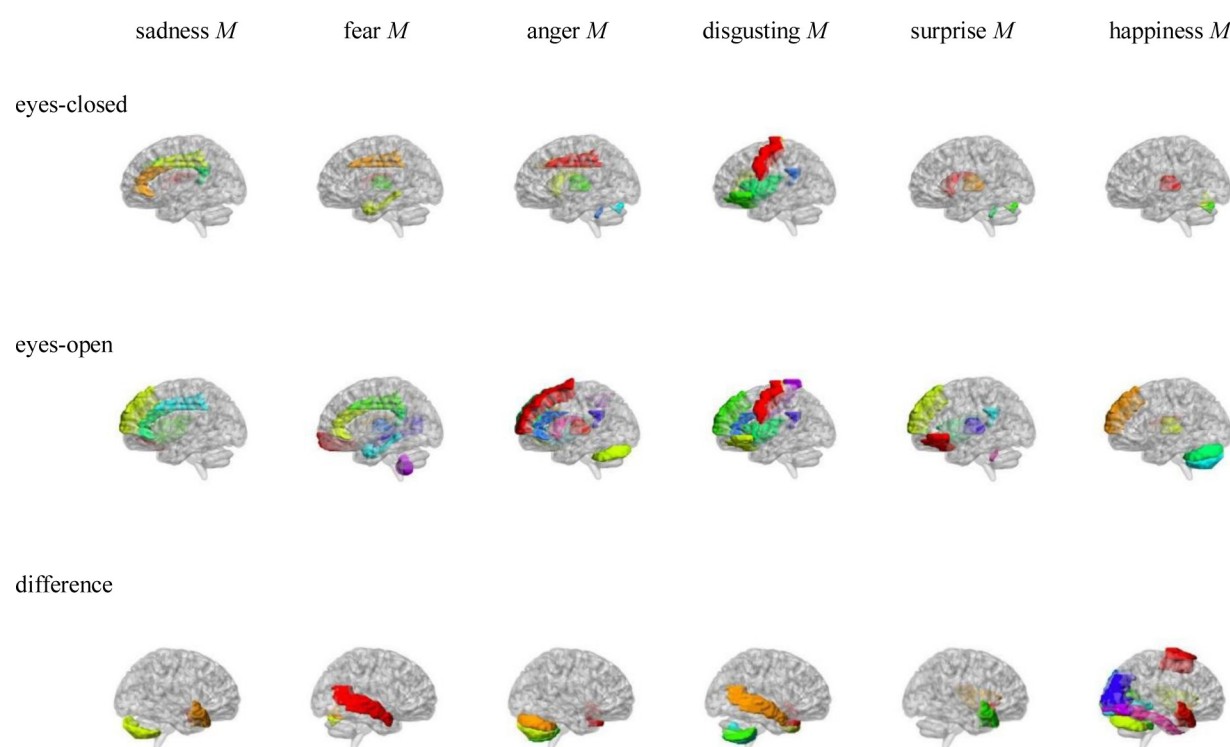

**Fig 2. AAL brain areas whose ALFFs in eyes-closed and eyes-open resting-state and ALFFs-difference were related to the microexpression *M*.** Note: The brain areas were visualised with the BrainNet Viewer (http://www.nitrc.org/projects/bnv/) [20], the same below.

almost all the microexpressions. Only fear under surprise was not significantly higher than random, which might because the fear microexpression and the surprise backgrounds had similar face muscle status. All the microexpression *SD*s were significantly higher than random, which means that each microexpression type had significantly background effect [6, 8].

Each microexpression *M* in the first EMERT was significantly positively related to the corresponding one in the second EMERT, and the *r*s were high. Each microexpression *SD* except surprise *SD* in the first EMERT was significantly positively related to the corresponding one in the second EMERT, which showed that the EMERT had good retest reliability. Each microexpression *M* in the first EMERT was significantly positively related to the corresponding microexpression in the first JACBART, which showed that the EMERT had good criterion validity [6, 8].

## 4.2 The relevant brain areas of microexpression *M* in EMERT

In the eyes-closed resting state, ALFFs in the frontal lobe, insula, cingulate cortex, hippocampal, caudate nucleus, thalamus and vermis were significantly correlated with some microexpression *M*, of which the insula, cingulate cortex, hippocampal and thalamus were common brain areas of expression recognition [21], the frontal lobe, insula, cingulate cortex, hippocampal and thalamus might be responsible for microexpressions consciousness and attention [22, 23], and the caudate nucleus and vermis might be responsible for the change from expression backgrounds to microexpression [10], which of course need further research to determine, the same as below.

In the eyes-open resting state, ALFFs in the frontal lobe, insula, cingulate cortex, hippocampus, parietal lobe, caudate nucleus, thalamus, temporal lobe, cerebellum and vermis were

**Table 4. The *r*s between ALFFs of resting-state and microexpression *SD*.**

| resting-state | AAL brain area | ALFF (*M±SD*) | sadness *SD* | fear *SD* | anger *SD* | disgust *SD* | surprise *SD* | happiness *SD* |
|---|---|---|---|---|---|---|---|---|
| eyes-closed | Frontal_Mid_R | 0.86±0.05 | -0.29* | | | | | |
| eyes-closed | Frontal_Inf_Tri_L | 0.86±0.04 | | 0.30* | | | | |
| eyes-closed | Frontal_Inf_Tri_R | 0.78±0.04 | | 0.26* | 0.27* | | | 0.26* |
| eyes-closed | Supp_Motor_Area_L | 1.02±0.07 | | -0.34** | | | | |
| eyes-closed | Supp_Motor_Area_R | 1.03±0.08 | | | | -0.28* | | |
| eyes-closed | Insula_L | 0.91±0.04 | | | | 0.37** | | |
| eyes-closed | Insula_R | 0.96±0.05 | | | | 0.28* | | |
| eyes-closed | Cingulum_Ant_R | 0.88±0.05 | -0.30* | | | | | |
| eyes-closed | Cingulum_Post_L | 0.98±0.05 | | | | 0.28* | | |
| eyes-closed | Calcarine_R | 1.01±0.1 | 0.31* | -0.26* | | | | |
| eyes-closed | Cuneus_R | 1.11±0.13 | 0.30* | | | | | |
| eyes-closed | Lingual_L | 1.07±0.1 | 0.37** | | | | | |
| eyes-closed | Lingual_R | 1.06±0.09 | 0.30* | | | | | |
| eyes-closed | Occipital_Sup_L | 0.92±0.08 | 0.33* | | | -0.28* | | |
| eyes-closed | Occipital_Sup_R | 0.89±0.06 | 0.33* | | | | | |
| eyes-closed | Occipital_Mid_L | 0.95±0.06 | 0.30* | | | | | |
| eyes-closed | Occipital_Inf_L | 0.92±0.07 | 0.27* | | | | | |
| eyes-closed | Occipital_Inf_R | 0.93±0.07 | 0.35** | | | | | |
| eyes-closed | Parietal_Inf_L | 0.99±0.05 | | | | -0.34** | | |
| eyes-closed | SupraMarginal_L | 0.89±0.04 | | 0.37** | | | | |
| eyes-closed | Precuneus_L | 1.08±0.05 | | -0.28* | | | | |
| eyes-closed | Caudate_L | 0.95±0.1 | | | -0.34* | | | |
| eyes-closed | Putamen_L | 0.79±0.04 | | | | 0.33* | | |
| eyes-closed | Pallidum_L | 0.83±0.05 | | | | 0.30* | | |
| eyes-closed | Thalamus_L | 1.01±0.09 | 0.31* | | | | | |
| eyes-closed | Thalamus_R | 1.01±0.08 | 0.31* | | | | 0.32* | |
| eyes-closed | Temporal_Pole_Sup_R | 1.02±0.08 | | | 0.36** | | | |
| eyes-closed | Cerebelum_3_R | 1.78±0.33 | | | | 0.28* | | |
| eyes-closed | Cerebelum_10_R | 0.99±0.17 | -0.43** | | | | | |
| eyes-closed | Vermis_4_5 | 1.21±0.14 | 0.33* | | -0.41** | | | |
| eyes-closed | Vermis_6 | 0.94±0.07 | | -0.31* | | | | |
| eyes-closed | Vermis_7 | 0.81±0.07 | | | | | 0.29* | |
| eyes-open | Frontal_Mid_Orb_L | 0.88±0.08 | 0.25* | | | | | |
| eyes-open | Rolandic_Oper_L | 0.84±0.03 | | | | 0.29* | -0.29* | |
| eyes-open | Supp_Motor_Area_L | 1±0.07 | 0.32* | | -0.25* | | | |
| eyes-open | Olfactory_L | 0.97±0.08 | | | -0.28* | | | |
| eyes-open | Frontal_Sup_Medial_L | 0.97±0.07 | | | -0.26* | | | -0.25* |
| eyes-open | Insula_L | 0.92±0.03 | | | | 0.43** | | |
| eyes-open | Insula_R | 0.97±0.04 | | | | 0.26* | | |
| eyes-open | Cingulum_Ant_L | 0.99±0.06 | | | -0.36** | | | |
| eyes-open | Cingulum_Ant_R | 0.89±0.04 | | | | 0.26* | | |
| eyes-open | Cingulum_Post_L | 1±0.05 | | 0.27* | | 0.29* | -0.41** | |
| eyes-open | Cingulum_Post_R | 0.94±0.03 | | 0.33** | | | | |
| eyes-open | Amygdala_L | 1.11±0.11 | | | | | -0.32* | |
| eyes-open | Amygdala_R | 1.07±0.1 | | | | | -0.26* | |
| eyes-open | Calcarine_R | 0.94±0.05 | 0.35** | -0.28* | | | | |
| eyes-open | Cuneus_L | 1.08±0.1 | | -0.31* | | | | |

(*Continued*)

**Table 4.** (Continued)

| resting-state | AAL brain area | ALFF (M±SD) | sadness SD | fear SD | anger SD | disgust SD | surprise SD | happiness SD |
|---|---|---|---|---|---|---|---|---|
| eyes-open | Cuneus_R | 1.05±0.07 | 0.35** | -0.39** | | | | |
| eyes-open | Lingual_L | 1.01±0.06 | 0.41** | | | | | |
| eyes-open | Parietal_Inf_L | 0.98±0.05 | | | | -0.26* | | |
| eyes-open | Angular_R | 1.06±0.08 | | | | -0.28* | | |
| eyes-open | Caudate_L | 0.96±0.09 | | | -0.30* | | | |
| eyes-open | Putamen_L | 0.8±0.03 | | | -0.26* | 0.25* | | |
| eyes-open | Putamen_R | 0.8±0.03 | | | | 0.26* | | |
| eyes-open | Pallidum_L | 0.84±0.04 | | | -0.30* | | | |
| eyes-open | Thalamus_L | 1±0.08 | 0.27* | | | | | |
| eyes-open | Thalamus_R | 1±0.07 | 0.28* | | | | 0.26* | |
| eyes-open | Cerebelum_Crus1_R | 0.97±0.1 | | -0.30* | | | | |
| eyes-open | Cerebelum_10_R | 0.99±0.21 | | | | | | 0.25* |
| eyes-open | Vermis_4_5 | 1.18±0.13 | | | -0.33** | | | |
| difference | Frontal_Mid_Orb_L | 0.04±0.07 | 0.31* | | | | | |
| difference | Frontal_Mid_Orb_R | 0.03±0.04 | 0.36** | | | -0.28* | | |
| difference | Frontal_Inf_Tri_R | 0.01±0.03 | 0.30* | | -0.281* | | | |
| difference | Frontal_Inf_Orb_L | 0.02±0.03 | | | | -0.27* | | |
| difference | Rolandic_Oper_L | 0±0.02 | | | | | | -0.26* |
| difference | Rolandic_Oper_R | -0.01±0.02 | | | -0.39** | | | |
| difference | Supp_Motor_Area_L | -0.02±0.04 | | | | 0.27* | | |
| difference | Supp_Motor_Area_R | -0.03±0.05 | | | | 0.28* | | |
| difference | Olfactory_L | 0.01±0.03 | | | | | -0.30* | |
| difference | Insula_R | 0.01±0.02 | | | -0.30* | | | |
| difference | Cingulum_Ant_R | 0.01±0.03 | 0.27* | | | | | |
| difference | Amygdala_R | 0.01±0.04 | | | -0.27* | | | |
| difference | Occipital_Sup_R | 0±0.05 | -0.30* | | | | | 0.28* |
| difference | Occipital_Inf_R | 0.01±0.06 | -0.32* | | | | | |
| difference | Fusiform_R | 0±0.02 | -0.31* | | | | | |
| difference | Paracentral_Lobule_R | -0.07±0.09 | | | | 0.32* | | |
| difference | Temporal_Pole_Sup_L | 0.01±0.05 | | | | | | -0.27* |
| difference | Temporal_Pole_Sup_R | 0±0.05 | | | | | -0.28* | |
| difference | Temporal_Pole_Mid_R | 0±0.03 | | | | | 0.30* | |
| difference | Temporal_Inf_R | 0.01±0.02 | | | -0.28* | | | |
| difference | Cerebelum_Crus1_R | -0.01±0.05 | | | | 0.28* | | |
| difference | Cerebelum_Crus2_L | -0.01±0.09 | | | | | | 0.30* |
| difference | Cerebelum_6_L | -0.02±0.05 | | | | | | 0.28* |
| difference | Cerebelum_7b_R | 0.01±0.1 | | | | -0.31* | | |
| difference | Cerebelum_8_R | 0.02±0.1 | | | | -0.29* | | |
| difference | Vermis_9 | -0.01±0.09 | | | 0.36** | | | |

significantly correlated with some microexpression $M$, of which the insula, cingulate cortex, hippocampus, thalamus and temporal lobe were common brain areas of expression recognition, the frontal lobe, parietal lobe, hippocampus, insula, cingulate, hippocampus, thalamus and temporal lobe might be responsible for microexpressions consciousness and attention. The caudate nucleus, cerebellum and vermis might be responsible for the change from expression backgrounds to microexpression [24]. It can be seen that microexpression $M$ was

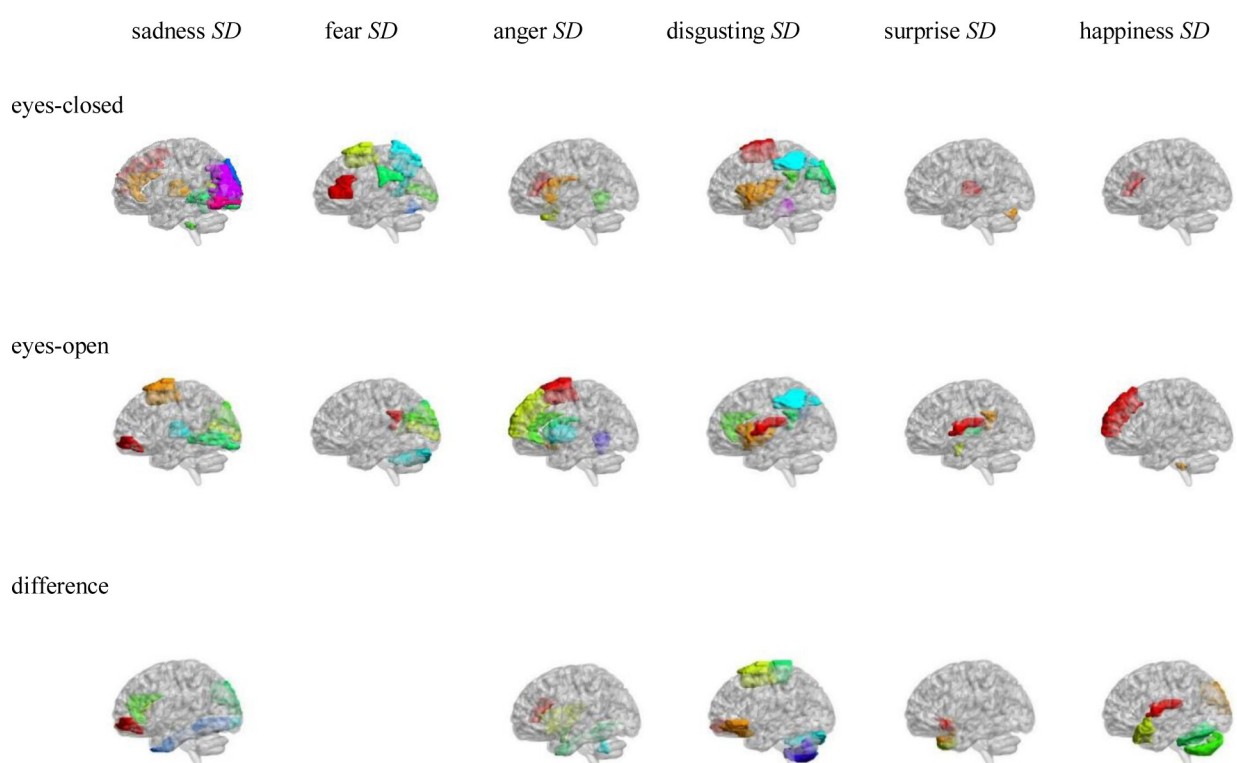

**Fig 3. AAL brain areas whose ALFFs in eyes-closed and eyes-open resting-state and ALFFs-difference were related to the microexpression SD.**

significantly correlated with similar brain areas in both eyes-closed and eyes-open resting-states. Still, in the eyes-open resting state, there were more relevant brain areas such as parietal lobe and temporal lobe.

In difference of eyes-open minus eyes-closed resting-states, ALFFs-difference in the frontal lobe, insula, amygdala, occipital lobe, fusiform, temporal lobe, cerebellum and vermis were significantly correlated with some microexpression $M$, of which the insula, amygdala, occipital lobe, fusiform and temporal lobe were common expression recognition brain areas, the frontal lobe, insula and temporal lobe might be responsible for microexpressions consciousness and attention. The cerebellum and vermis might be responsible for the change from expression backgrounds to microexpression. It can be seen that there were some similar relevant brain areas in the ALFFs-difference as in eyes-closed and eyes-open resting-states. Still, in the ALFFs-difference, there were new relevan brain areas such as the amygdala, occipital lobe and fusiform.

It was found that ALFFs in both eyes-closed and eyes-open resting-states and ALFFs-difference could predict microexpression $M$ of EMERT. Their predictability was similar, but there were also differences. According to the relevant brain areas and logic, there might be three cognitive processes in ecological microexpression recognition, such as the expression recognition, microexpressions consciousness and attention, and the change from expression background to microexpression. We need to explore whether and when each of them occurs and whether some other cognitive processes exist by developing new behavioural measurement methods to separate them and by task-state fMRI and ERP in the future. Nakano et al. [15, 16] found that transition from eyes-closed to eyes-open was from internal feeling to external stimulus processing. However, no study has taken the ALFFs-difference as a quantitative sensitivity index

from internal feeling to external stimulus, and no study has investigated its psychological significance. In the current study, we defined the ALFFs-difference as the quantitative sensitivity index from internal feeling to external stimulus, and it was found that the ALFFs-difference could predict EMERT, indicating psychological significance.

Shen [9] found that the brain area responsible for classical microexpression recognition was in the left frontal lobe, while the brain area responsible for expression recognition was in the right frontal lobe. In the current study, it was found that for EMERT, both the left and right frontal lobes and more brain areas were involved. Zhang [10] found that for anger and neutral microexpressions, activation of the inferior parietal lobule was induced more in the negative expression backgrounds than in the neutral expression backgrounds, while activation of the right precuneus was induced more in the positive expression backgrounds than in the neutral expression backgrounds. For happiness microexpressions, activation of the parahippocampal gyrus was induced more in the positive backgrounds. The current study also found that these brain areas except the right precuneus were involved in EMERT, and more brain areas were involved. There might be three reasons for this difference: (1) The EMERT in the current study was more comprehensive and ecological, and there was more background effect. (2) The correlation analysis of resting-state was adopted in the current study. Still, the comparative analysis of task-states has been used in previous studies either between microexpressions and expressions or among different microexpressions, therefore many common brain areas either of microexpressions and expressions or of different microexpressions might be ignored by statistics. (3) We detected the relevant brain areas of new recapitulative index of EMERT such as microexpression *M* and did not pay attention to details. Zhang et al. [6] established EMERT but did not investigate the relevant brain areas. In the current study, the relevant brain areas of EMERT were comprehensively investigated. Of course, further researches are needed to determine which function brain areas are responsible for.

## 4.3 The relevant brain areas of microexpression *SD* in EMERT

In the eyes-closed resting-state, ALFFs in the frontal lobe, insula, cingulate cortex, occipital lobe, parietal lobe, precuneus, caudate lobe, putamen lobe, thalamus, temporal lobe, cerebellum and vermis were significantly correlated with some microexpression *SD*. In the eyes-open resting-state, ALFFs in the frontal lobe, insula, cingulate cortex, cuneus, occipital lobe, parietal lobe, precuneus, caudate lobe, putamen lobe, thalamus, temporal lobe, cerebellum and vermis were significantly correlated with some microexpression *SD*. It can be seen that microexpression *SD* was significantly associated with similar brain areas in both eyes-closed and eyes-open resting-states.

In the ALFFs-difference of eyes-open minus eyes-closed resting-states, ALFFs-difference in the frontal lobe, insula, cingulate cortex, amygdala, fusiform, occipital lobe, parietal lobe, temporal lobe, cerebellum and vermis were significantly correlated with some microexpression *SD*. It can be seen that there were many similar relevant brain areas in the ALFFs-difference as in eyes-closed and eyes-open resting-states. Still, in the ALFFs-difference, there were new relevant brain areas such as amygdala and fusiform.

It was found that ALFFs in both eyes-closed and eyes-open resting-states and ALFFs-difference could predict microexpression *SD*. Their predictability was similar, but there were also differences. In EMERT, Zhang et al. [6] and Yin, Tian, Hua, Zhang, & Liu [8] defined the microexpression *SD* as the fluctuation of the ecological micro expression to quantify the background effect. Still, they did not investigate the relevant brain areas. The current study comprehensively investigated the relevant brain areas involved in the quantification of the background effect. Of course, further researches are needed to determine which function brain areas are responsible for.

### 4.4 The similarities and differences of the relevant brain areas of microexpression *M* and *SD*

The microexpression *M* is the index of a microexpression type recognition. The microexpression *SD* is the background effect index of this microexpression type recognition [6, 8]. The former is a kind of ability, but the latter is the degree that this ability changes in different contexts, which, in turn, can be thought of as the stability of this ability. Therefore, there should be similarities and differences in brain mechanisms between them.

In the eyes-closed resting state, ALFFs in the frontal lobe, insula, cingulate cortex, caudate nucleus, thalamus and vermis were significantly correlated with both some microexpression *M* and some microexpression *SD*, which indicates they need emotional perception and feeling. But ALFFs in hippocampal were only significantly correlated with both some microexpression *M*, which indicates that the microexpression type recognition ability need memory more; and ALFFs in the occipital lobe, parietal lobe, precuneus, putamen lobe, temporal lobe and cerebellum were only significantly correlated with some microexpression *SD*, which indicates that the stability of the microexpression type recognition ability need cognitive control, consciousness and motion more.

In the eyes-open resting state, ALFFs in the frontal lobe, insula, cingulate cortex, parietal lobe, caudate nucleus, thalamus, temporal lobe, cerebellum and vermis were significantly correlated with both some microexpression *M* and some microexpression *SD*, which indicates they need emotional perception and feeling. But ALFFs in hippocampal were only significantly correlated with both some microexpression *M*, which indicates that the microexpression type recognition ability need memory more; and ALFFs cuneus, occipital lobe, precuneus and putamen lobe were significantly correlated with some microexpression *SD*, which indicates that the stability of the microexpression type recognition ability need visual, consciousness and motion more.

In difference of eyes-open minus eyes-closed resting-states, ALFFs-difference in the frontal lobe, insula, amygdala, occipital lobe, fusiform, temporal lobe, cerebellum and vermis were significantly correlated with both some microexpression *M* and some microexpression *SD*, which indicates they need emotional perception and feeling. But ALFFs-difference in the cingulate cortex and parietal lobe was only significantly correlated with some microexpression *SD*, which indicates that the stability of the microexpression type recognition ability needs cognitive control more.

Taken together, both microexpression *M* and microexpression *SD* need emotional perception and feeling, but the former need memory more, and the latter need cognitive control and consciousness more. Of course, a certain ability requires its related brain areas and memory, but in addition to the brain areas associated with this ability, the stability of this ability requires cognitive control and consciousness. The similarities and differences in brain mechanisms of microexpression *M* and *SD* are logical. All these relevant brain areas can be trained to enhance ecological microexpression recognition ability.

## 5 Conclusion

The current study used white, black and yellow models' expressions as microexpressions and backgrounds to improve the materials ecological validity of EMERT. It used eyes-closed and eyes-open resting-state fMRI to detect relevant resting-state brain activity of EMERT. The result showed:

1. Two new recapitulative indexes of EMERT were adopted, such as microexpression *M* and microexpression *SD*. The participants could effectively identify almost all the

microexpressions, and each microexpression type had a significantly background effect. The EMERT had good retest reliability and calibration validity.

2. ALFFs in both eyes-closed and eyes-open resting-states and ALFFs-difference could predict microexpression *M*. The relevant brain areas of microexpression *M* were some frontal lobes, insula, cingulate cortex, hippocampus, parietal lobe, caudate nucleus, thalamus, amygdala, occipital lobe, fusiform, temporal lobe, cerebellum and vermis.

3. ALFFs in both eyes-closed and eyes-open resting-states and ALFFs-difference could predict microexpression *SD*, and the ALFFs-difference was more predictive. The relevant brain areas of microexpression *SD* were some frontal lobes, insula, cingulate cortex, cuneus, amygdala, fusiform, occipital lobe, parietal lobe, precuneus, caudate lobe, putamen lobe, thalamus, temporal lobe, cerebellum and vermis.

4. There were many similarities and some differences in the relevant brain areas between microexpression *M* and *SD*. All these brain areas can be trained to enhance ecological microexpression recognition ability.

## Supporting information

**S1 File.**
(SAV)

## Author Contributions

**Conceptualization:** Ming Yin, Jianxin Zhang.

**Data curation:** Jianxin Zhang.

**Formal analysis:** Ming Yin, Jianxin Zhang.

**Funding acquisition:** Ming Yin, Jianxin Zhang, Dianzhi Liu.

**Investigation:** Jianxin Zhang.

**Methodology:** Jianxin Zhang, Deming Shu, Dianzhi Liu.

**Project administration:** Dianzhi Liu.

**Resources:** Dianzhi Liu.

**Software:** Deming Shu.

**Supervision:** Dianzhi Liu.

**Visualization:** Jianxin Zhang.

**Writing – original draft:** Jianxin Zhang.

**Writing – review & editing:** Ming Yin, Jianxin Zhang, Dianzhi Liu.

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
