## [Decision Letter · Decision Letter 0]

9 Jul 2020

PONE-D-20-03018

The relevant resting-state brain activity of ecological microexpression recognition test (EMERT) ☆

PLOS ONE

Dear Dr. Zhang,

Thank you for submitting your manuscript to PLOS ONE. After careful consideration, we feel that it has merit but does not fully meet PLOS ONE’s publication criteria as it currently stands. Therefore, we invite you to submit a revised version of the manuscript that addresses the points raised during the review process.

We look forward to receiving your revised manuscript.

Kind regards,

Kai Wang

Academic Editor

PLOS ONE

Journal Requirements:

2. Please ensure that you have included a detailed section on the statistical analysis you performed in this study in your Methods section. Please ensure that you have indicated how you corrected for multiple comparisons. For further guidance please see here: https://journals.plos.org/plosone/s/submission-guidelines.#loc-statistical-reporting.

3. Please provide additional details regarding participant consent. In the Methods section, please ensure that you have specified what type of consent you obtained (for instance, written or verbal) and whether the ethics committee approved this consent procedure. If verbal consent was obtained please state why it was not possible to obtain written consent and how verbal consent was recorded. If your study included minors, state whether you obtained consent from parents or guardians.

6. We note that Figure 1 includes an image of a [patient / participant / in the study]. 

Additional Editor Comments (if provided):

Reviewers' comments:

Reviewer's Responses to Questions

**Comments to the Author**

1. Is the manuscript technically sound, and do the data support the conclusions?

Reviewer #1: Partly

Reviewer #2: Yes

2. Has the statistical analysis been performed appropriately and rigorously? 

Reviewer #1: Yes

Reviewer #2: Yes

3. Have the authors made all data underlying the findings in their manuscript fully available?

Reviewer #1: Yes

Reviewer #2: Yes

4. Is the manuscript presented in an intelligible fashion and written in standard English?

Reviewer #1: Yes

Reviewer #2: Yes

5. Review Comments to the Author

Reviewer #1: This manuscript used white, black and yellow models’ expressions as microexpressions and backgrounds to improve the materials ecological validity of EMERT, and used eyes-closed and eye-open resting-state fMRI to detect relevant brain activity of EMERT for the first time. The main findings demonstrated there were many similarities and some differences of the relevant brain areas between microexpression M and SD.

I have several concerns that need to be addressed.

Main issues:

-About the EMERT’s retest reliability, which experimental results demonstrated that the EMERT had good retest reliability.

-The similarities and differences of the relevant brain areas of microexpression M and SD were not detailedly discussed in “Discussion” section, which tells readers less information.

-Do you have any criterion to choose the types of basic microexpressions or emotion model.

Minor issues:

-On page 5, need a reference for head motion correction.

-On page 7, is there any error for “A ingle sample t test”.

-On page 7, what does “rs” mean.

Reviewer #2: I just three minor questions. One is that the ALFF in abstract should be provided the full name for the first time. The second one is that the referrences should be carefully checked. Third, the language should be edited by a native speaker.

6. PLOS authors have the option to publish the peer review history of their article (what does this mean?). If published, this will include your full peer review and any attached files.

Reviewer #1: No

Reviewer #2: No

---

## [Author Response · Author response to Decision Letter 0]

3 Sep 2020

Reviewer #1: This manuscript used white, black and yellow models’ expressions as microexpressions and backgrounds to improve the materials ecological validity of EMERT, and used eyes-closed and eyes-open resting-state fMRI to detect relevant brain activity of EMERT for the first time. The main findings demonstrated there were many similarities and some differences of the relevant brain areas between microexpression M and SD.

I have several concerns that need to be addressed.

Answer: Thank you very much for your advice. We have revised the article according to your comments. Please see the blue font in the revised manuscript for details. 

Main issues:

-About the EMERT’s retest reliability, which experimental results demonstrated that the EMERT had good retest reliability.

Answer: In the current study, the same participants were asked to do the same EMERT twice. Therefore the EMERT’s retest reliability could be measured by Pearson correlation between the two EMERT. It was found that each microexpression M in the first EMERT was significantly positively related to the corresponding one in the second EMERT and the rs (the plural of r, the same below) were high; and that each microexpression SD except surprise SD in the first EMERT was significantly positively related to the corresponding one in the second EMERT. These results showed that the EMERT had good retest reliability. 

-The similarities and differences of the relevant brain areas of microexpression M and SD were not detailedly discussed in “Discussion” section, which tells readers less information.

Answer: Thank you very much! This advice was instructive. We have added the specific similarities and differences of the relevant brain areas of microexpression M and SD. Please see the blue font in 4.4 for details. 

The microexpression M is the index of a microexpression type recognition. The microexpression SD is the background effect index of this microexpression type recognition (Yin, Tian, Hua, Zhang, & Liu, 2019; Zhang et al., 2017). The former is a kind of ability, but the latter is the degree that this ability changes in different contexts, which, in turn, can be thought of as the stability of this ability. Therefore, there should be similarities and differences in brain mechanisms between them.

In the eyes-closed resting state, ALFFs in the frontal lobe, insula, cingulate cortex, caudate nucleus, thalamus and vermis were significantly correlated with both some microexpression M and some microexpression SD, which indicates they need emotional perception and feeling. But ALFFs in hippocampal were only significantly correlated with both some microexpression M, which indicates that the microexpression type recognition ability need memory more; and ALFFs in the occipital lobe, parietal lobe, precuneus, putamen lobe, temporal lobe and cerebellum were only significantly correlated with some microexpression SD, which indicates that the stability of the microexpression type recognition ability need cognitive control, consciousness and motion more. 

In the eyes-open resting state, ALFFs in the frontal lobe, insula, cingulate cortex, parietal lobe, caudate nucleus, thalamus, temporal lobe, cerebellum and vermis were significantly correlated with both some microexpression M and some microexpression SD, which indicates they need emotional perception and feeling. But ALFFs in hippocampal were only significantly correlated with both some microexpression M, which indicates that the microexpression type recognition ability need memory more; and ALFFs cuneus, occipital lobe, precuneus and putamen lobe were significantly correlated with some microexpression SD, which indicates that the stability of the microexpression type recognition ability need visual, consciousness and motion more. 

In difference of eyes-open minus eyes-closed resting-states, ALFFs-difference in the frontal lobe, insula, amygdala, occipital lobe, fusiform, temporal lobe, cerebellum and vermis were significantly correlated with both some microexpression M and some microexpression SD, which indicates they need emotional perception and feeling. But ALFFs-difference in the cingulate cortex and parietal lobe was only significantly correlated with some microexpression SD, which indicates that the stability of the microexpression type recognition ability needs cognitive control more. 

Taken together, both microexpression M and microexpression SD need emotional perception and feeling, but the former need memory more, and the latter need cognitive control and consciousness more. Of course, a certain ability requires its related brain areas and memory, but in addition to the brain areas associated with this ability, the stability of this ability requires cognitive control and consciousness. The similarities and differences in brain mechanisms of microexpression M and SD are logical. All these relevant brain areas can be trained to enhance ecological microexpression recognition ability. 

-Do you have any criterion to choose the types of basic microexpressions or emotion model.

Answer: We chose the same six types of basic microexpressions as EMERT (Zhang et al., 2017), but the models must contain white, black and yellow people to improve the degree of ecologization for EMERT. So the criterion was as follows: Seven kinds of basic expression opened mouth pictures of eight models (four male and four female, including white, black and yellow people) from the NimStim face expression database (Tottenham et al., 2009) were used as the backgrounds, namely, neutral, sadness, fear, anger, disgust, surprise, and happiness. Except for neutral expression, the other six kinds of expressions were used as microexpressions. The pixels of all images were modified to be 338 × 434 with grey background (GRB: 127, 127, 127) (Zhang et al., 2017). Since each model in the NimStim face expression database had only two images for each expression, such as mouth-open and mouth-closed, we chose mouth-open with stable pleasure and arousal, so we were unable to measure different pleasure and arousal of a microexpression.

Minor issues:

-On page 5, need a reference for head motion correction.

-On page 7, is there any error for “A ingle sample t test”.

-On page 7, what does “rs” mean.

Answer: 

-On page 5, a reference for head motion correction (adopting Friston 24) was added.

-On page 7, the correct spelling was “A single sample t test”.

-On page 7, “rs” means the plural of r, the same below.

Reviewer #2: I just three minor questions. One is that the ALFF in abstract should be provided the full name for the first time. The second one is that the referrences should be carefully checked. Third, the language should be edited by a native speaker.

Answer: Thank you very much for your advice. We have revised the article according to your comments. Please see the blue font in the revised manuscript for details. The ALFF in abstract was provided the full name “Amplitude of Low-Frequency Fluctuations” for the first time. The referrences were carefully checked and we added one referrence. The language was edited by a translation company.

---

## [Decision Letter · Decision Letter 1]

20 Oct 2020

The relevant resting-state brain activity of ecological microexpression recognition test (EMERT) ☆

PONE-D-20-03018R1

Dear Dr. Zhang,

We’re pleased to inform you that your manuscript has been judged scientifically suitable for publication and will be formally accepted for publication once it meets all outstanding technical requirements.

Kind regards,

Kai Wang

Academic Editor

PLOS ONE

Additional Editor Comments (optional):

Reviewers' comments:

Reviewer's Responses to Questions

**Comments to the Author**

1. If the authors have adequately addressed your comments raised in a previous round of review and you feel that this manuscript is now acceptable for publication, you may indicate that here to bypass the “Comments to the Author” section, enter your conflict of interest statement in the “Confidential to Editor” section, and submit your "Accept" recommendation.

Reviewer #1: All comments have been addressed

Reviewer #2: All comments have been addressed

2. Is the manuscript technically sound, and do the data support the conclusions?

Reviewer #1: Yes

Reviewer #2: Yes

3. Has the statistical analysis been performed appropriately and rigorously? 

Reviewer #1: Yes

Reviewer #2: Yes

4. Have the authors made all data underlying the findings in their manuscript fully available?

Reviewer #1: Yes

Reviewer #2: No

5. Is the manuscript presented in an intelligible fashion and written in standard English?

Reviewer #1: Yes

Reviewer #2: Yes

6. Review Comments to the Author

Reviewer #1: All conerns have been addressed by authors in the submitted manuscript. So, I have no further questions.

Reviewer #2: (No Response)

7. PLOS authors have the option to publish the peer review history of their article (what does this mean?). If published, this will include your full peer review and any attached files.

Reviewer #1: No

Reviewer #2: No

---

## [Editor Report · Acceptance letter]

13 Nov 2020

PONE-D-20-03018R1 

The relevant resting-state brain activity of ecological microexpression recognition test (EMERT) ☆ 

Dear Dr. Zhang:

I'm pleased to inform you that your manuscript has been deemed suitable for publication in PLOS ONE. Congratulations! Your manuscript is now with our production department. 

Kind regards, 

on behalf of

Prof. Kai Wang 

Academic Editor

PLOS ONE